# CyberKnife in Pediatric Oncology: A Narrative Review of Treatment Approaches and Outcomes

**DOI:** 10.3390/curroncol32020076

**Published:** 2025-01-29

**Authors:** Costanza M. Donati, Federica Medici, Arina A. Zamfir, Erika Galietta, Silvia Cammelli, Milly Buwenge, Riccardo Masetti, Arcangelo Prete, Lidia Strigari, Ludovica Forlani, Elisa D’Angelo, Alessio G. Morganti

**Affiliations:** 1Radiation Oncology, IRCCS Azienda Ospedaliero-Universitaria di Bologna, Via Albertoni 15, 40128 Bologna, Italy; costanzamaria.donati@unibo.it (C.M.D.); erika.galietta2@unibo.it (E.G.); silvia.cammelli2@unibo.it (S.C.); amorganti60@gmail.com (A.G.M.); 2Department of Medical and Surgical Sciences (DIMEC), Alma Mater Studiorum University of Bologna, 40100 Bologna, Italy; milly.buwenge@unibo.it (M.B.); riccardo.masetti5@unibo.it (R.M.); arcangelo.prete@aosp.bo.it (A.P.); ludovica.forlani@studio.unibo.it (L.F.); 3Département de Radiothérapie, Gustave Roussy, 94805 Villejuif, France; federica.medici4@gmail.com; 4Pediatric Oncology and Hematology “Lalla Seràgnoli”, IRCCS Azienda Ospedaliero Universitaria di Bologna, 40128 Bologna, Italy; 5Department of Medical Physics, IRCCS Azienda Ospedaliero-Universitaria di Bologna, 40128 Bologna, Italy; lidia.strigari2@unibo.it; 6Department of Radiation Oncology, Bellaria Hospital-AUSL Bologna, 40139 Bologna, Italy; elisa.dangelo5@unibo.it

**Keywords:** literature review, narrative review, robotic accelerator, CyberKnife, pediatric, radiotherapy

## Abstract

Pediatric cancers, while rare, pose unique challenges due to the heightened sensitivity of developing tissues and the increased risk of long-term radiation-induced effects. Radiotherapy (RT) is a cornerstone in pediatric oncology, but its application is limited by concerns about toxicity, particularly secondary malignancies, growth abnormalities, and cognitive deficits. CyberKnife (CK), an advanced robotic radiosurgery system, has emerged as a promising alternative due to its precision, non-invasiveness, and ability to deliver hypofractionated, high-dose RT while sparing healthy tissues. This narrative review explores the existing evidence on CK application in pediatric patients, synthesizing data from case reports, small series, and larger cohort studies. All the studies analyzed reported cases of tumors located in the skull or in the head and neck region. Findings suggest CK’s potential for effective tumor control with favorable toxicity profiles, especially for complex or inoperable tumors. However, the evidence remains limited, with the majority of studies involving small sample sizes and short follow-up periods. Moreover, concerns about the “dose-bath” effect and limited long-term data on stochastic risks warrant cautious adoption. Compared to Linac-based RT and proton therapy, CK offers unique advantages in reducing session numbers and enhancing patient comfort, while its real-time tracking provides superior accuracy. Despite these advantages, CK is associated with significant limitations, including a higher potential for low-dose scatter (often referred to as the “dose-bath” effect), extended treatment times in some protocols, and high costs requiring specialized expertise for operation. Emerging modalities like π radiotherapy further underscore the need for comparative studies to identify the optimal technique for specific pediatric cases. Notably, proton therapy remains the benchmark for minimizing long-term toxicity, but its cost and availability limit its accessibility. This review emphasizes the need for balanced evaluations of CK and highlights the importance of planning prospective studies and long-term follow-ups to refine its role in pediatric oncology. A recent German initiative to establish a CK registry for pediatric CNS lesions holds significant promise for advancing evidence-based applications and optimizing treatment strategies in this vulnerable population.

## 1. Introduction

Pediatric cancers, though rare, present unique challenges in terms of treatment due to their aggressive nature and different tolerance to therapy, compared to adults, because of dissimilar host characteristics, such as physiology and organ maturation [1]. Among the available treatment modalities, radiotherapy (RT) has demonstrated efficacy in tumor control. However, despite recent technological advances [2], its application in pediatric patients is often limited due to concerns about potential long-term toxicity in developing tissues. In fact, delivering RT to pediatric patients requires extreme caution due to several factors: (i) increased sensitivity to radiation of children’s developing tissues and organs; (ii) longer life expectancy and therefore more time for radiation-induced late effects to manifest; (iii) risk of developing secondary cancers later in life; and (iv) risk of growth abnormalities and cognitive deficits [3,4,5,6].

CyberKnife (CK), also known as robotic radiosurgery or frameless radiosurgery, is an advanced and precise system for delivering high-dose radiation to tumors. Unlike conventional RT, CK employs a robotic arm to maneuver the treatment delivery, enabling exceptional targeting accuracy and real-time tracking of tumor movement during treatment. This frameless system utilizes non-invasive image-guided localization and a lightweight high-energy radiation source to deliver stereotactic radiosurgery in single or multiple sessions, often referred to as “ultra-hypofractionated” treatments typically involving two to five fractions, allowing ablative radiosurgical doses to the lesion while enhancing protection of adjacent tissues (Figure 1A,B).

CK is equipped with sophisticated image guidance technologies, enabling precise tumor localization within the body. During treatment setup, patients are immobilized using a custom-fitted mask, and in-room lasers define the center of the imaging system for initial alignment. The treatment location system employs orthogonal kV X-ray pairs, or live images, to compare the patient’s position against planning system-generated digitally reconstructed radiographs from the planning CT scan. This ensures alignment to within a millimeter of the planned treatment site. Additionally, the robotic couch performs fine adjustments in translation and rotation, including pitch, roll, and yaw, until residual offsets are within acceptable thresholds (<1 mm in translation and <0.5° in rotation). These offsets are continuously monitored and corrected during treatment.

CK’s frameless design eliminates the need for rigid head fixation, making it particularly suitable for treating tumors in challenging locations, such as the brain and spine. Furthermore, the system can perform real-time positional adjustments to account for any intrafraction movement, ensuring sub-millimeter precision throughout treatment. This combination of advanced tracking, non-invasiveness, and high precision positions CK as an effective and versatile tool for treating various tumor types while maintaining an enhanced focus on patient safety and comfort [7].

This high precision in treatment delivery, particularly in the capacity to minimize radiation exposure to healthy organs, especially those that are radiosensitive during their development, makes CK an appealing therapeutic choice in the pediatric population.

Despite these advantages, CK is associated with significant limitations, including a higher potential for low-dose scatter (often referred to as the “dose-bath” effect) and extended treatment times in some protocols, particularly when compared to other modalities such as Gamma Knife or proton therapy. Additionally, the high cost of CK systems and the specialized expertise required for their operation further restrict their widespread adoption, especially in resource-limited settings.

Moreover, while experiences with CK in adult patients have shown promising outcomes, including improved tumor control and reduced toxicity [8], the use of CK in the pediatric setting remains relatively unexplored, with only a few reports available in the literature predominantly consisting of small case series and retrospective reviews [9,10,11,12,13,14,15,16,17,18,19,20,21]. Specifically, the literature is deficient in reports on prospective studies, as well as in reviews of the existing evidence.

The present review seeks to address this gap by critically evaluating the existing literature on CK application in pediatric patients. In doing so, the review aims to offer a balanced perspective, acknowledging both the potential advantages and significant challenges of CK in this sensitive population. Another aim of this review is to explore and discuss possible comparisons between the results of CK and those of other established techniques (such as image-guided RT, proton therapy, intensity-modulated proton therapy, and Gamma Knife) or emerging techniques (such as 4π RT), specifically within the clinical settings of pediatric tumors for which CK is intended, namely small tumor lesions treated with few fractions.

## 2. Materials and Methods

### 2.1. Inclusion and Exclusion Criteria

This narrative review focused exclusively on studies that investigated the use of CK in pediatric patients. The review was conducted by a multidisciplinary team, composed of radiation oncologists, pediatric oncologists, and medical physicists, based on the Scale for the Assessment of Narrative Review Articles (SANRA) guidelines [22]. Abstracts, letters, editorials, and papers not written in English were excluded from the review.

### 2.2. Literature Search

A literature search was performed on PubMed on 13 July 13 2023, using the following search strategy: (“cyberknife” AND (“pediatric” OR “paediatric” OR “children”)) with filters applied: “Child: birth-18 years”. Additionally, the snowball technique was utilized to identify relevant articles by manually reviewing the reference lists of retrieved studies.

### 2.3. Study Selection and Data Extraction

Two authors (CMD, FM) independently screened the titles and abstracts of the identified articles to determine their relevance to the topic. Any disagreements between the authors were resolved through discussion and consultation with the senior author (AGM). Only studies meeting the inclusion criteria were considered for further analysis. Two authors (MB, SCa) independently extracted relevant information from the selected studies, including study characteristics, patient demographics, treatment protocols, tumor response, and toxicity outcomes. Any discrepancies or conflicts in data extraction were resolved through discussion and consensus.

### 2.4. Narrative Review Checklist

In order to ensure a thorough and comprehensive review of the topic, we adhered to a narrative review checklist. Appendix A outlines the checklist items and their corresponding assessment criteria, which guided our approach. By following this methodology, our objective was to conduct a comprehensive exploration of the literature pertaining to the application of CK in pediatric oncology.

## 3. Results

The initial search yielded a total of 59 items. After screening the titles and abstracts, 13 papers were identified as meeting the inclusion criteria, while 46 papers were excluded. All included papers were published since 2000, with 7 of them published since 2015, indicating a recent focus on the topic. The publications were sourced from various centers worldwide, including the USA (5 papers), Japan (4 papers), Costa Rica (1 paper), France (1 paper), UK (1 paper), and Turkey (1 paper). The selected papers covered a range of clinical settings, including brain tumors [11,12,17,19,21], oculomotor schwannomas [18], craniopharyngiomas [17], ameloblastic fibro-odontosarcoma [15], optic nerve glioma [14], clear cell meningioma [13], acoustic schwannoma [9], and juvenile nasopharyngeal angiofibroma [10]. However, all the cases and series analyzed focused on the treatment of tumors located in the intracranial or head and neck region. A summary of the findings of the selected studies is presented in Table 1.

### Literature Review

The application of CK in pediatric oncology, as reflected in the current literature, spans a spectrum of tumor types and treatment scenarios.

Case reports and small series [9,10,12,13,14,15,16,17,18,21] underscore CK’s capacity for precision in targeting a range of pediatric tumors, including but not limited to acoustic schwannoma, brain metastases, recurrent medulloblastoma, nasopharyngeal angiofibroma, optic gliomas, and clear cell meningioma. These studies collectively highlight CK’s potential advantages in minimizing radiation exposure to non-target tissues. However, the nature of these reports limits their ability to inform robust conclusions due to the absence of control groups, small sample sizes, and often short follow-up periods, which are insufficient for assessing long-term outcomes and late-onset toxicities. Additionally, these studies do not provide sufficient data on the comparative effectiveness of CK versus other high-precision techniques like Gamma Knife, raising questions about its specific niche in pediatric oncology.

Giller et al.’s experience [11] with 21 children with central nervous system tumors and Mohamad et al.’s study [19] involving 52 pediatric patients with brain tumors provide a broader perspective on CK utility, demonstrating notable success in achieving local control with minimal immediate adverse effects. These larger cohort studies contribute valuable insights into CK’s efficacy, yet still leave questions regarding the long-term safety profile and risk of secondary malignancies largely unanswered. For instance, the lack of comprehensive follow-up data on secondary malignancies remains a critical gap, particularly given CK’s association with a higher “dose-bath” effect compared to other technologies.

In the first paper [11], the authors discussed their experience using CK radiosurgery in the treatment of 21 pediatric patients with unresectable tumors. A total of 38 procedures were performed on children aged between 8 months and 16 years (average age 7 years). The tumors treated included pilocytic astrocytomas (3 cases), anaplastic astrocytomas (2 cases), ependymomas (3 cases, including 2 anaplastic), medulloblastomas (4 cases), atypical teratoid/rhabdoid tumors (3 cases), craniopharyngiomas (3 cases), and other pathologies (3 cases). The average target volume was 10.7 cm^3^, with a mean marginal dose of 18.8 Gy, and the follow-up period averaged 18 months. Of the procedures, 71% were single-session treatments, and 38% of patients did not require general anesthesia. Results indicated successful local control for patients with pilocytic and anaplastic astrocytomas, three patients with medulloblastomas, and all patients with craniopharyngiomas. However, local control was not achieved for those with ependymomas. Two patients with atypical teratoid/rhabdoid tumors survived for 16 and 35 months post-diagnosis. Notably, there were no deaths or complications related to the procedures. The authors concluded that CK radiosurgery proved effective in achieving local control for certain pediatric CNS tumors, without the need for rigid head fixation [11]. While these results are promising, the study’s small sample size and short follow-up highlight the need for further investigation into long-term outcomes and broader applicability.

The authors of the second paper [19] treated 52 pediatric brain tumor patients using CK stereotactic RT with doses of 1.8 to 2 Gy per fraction between 2008 and 2017. They compared thirty cases with intensity-modulated RT plans and assessed normal tissue exposure, plan quality, and dose–volume parameters, as well as overall survival, progression-free survival, and local control. Results indicated that CK plans exposed significantly less normal tissue to high doses (defined as ≥80% of the prescription dose or ≥40 Gy) and intermediate doses (defined as 80% > dose ≥ 50% of the prescription dose or 40 Gy > dose ≥ 25 Gy) compared to IMRT plans. With a median follow-up of 3.7 years, the 3-year local control rate was 92%. There were eight treatment failures: one craniopharyngioma, two ependymomas, and five low-grade gliomas. The authors concluded that CK SRT reduces the volume of irradiated tissue without significantly compromising local control in pediatric brain tumors, suggesting the need for further validation in prospective studies [19]. However, the study does not address the impact on broader clinical decision-making, particularly in the context of other available high-precision modalities. Furthermore, the authors of this study did not report data on the volume irradiated at low doses, which may be correlated with the incidence of secondary malignancies.

The review of Fadel et al. [18] offers a deeper dive into the technical merits and potential of CK, especially with respect to non-isocentric planning and the treatment of complex tumor geometries. These contributions emphasize the technological advancements that facilitate the tailored application of CK in pediatric cases, suggesting an improvement in radiation dose distribution and a theoretical reduction in harm to surrounding healthy tissue. Nevertheless, these technical advantages must be weighed against practical challenges such as longer treatment durations, higher costs, and the need for specialized expertise.

Overall, despite their limitations, the selected studies suggest that CK has a favorable toxicity profile for both acute and late deterministic effects. However, this observation applies strictly to the specific clinical settings for which this technique is designed and potentially effective (namely, small tumor lesions treated with few fractions) and must be interpreted with caution given the absence of comparative studies. Among the 98 patients included in the selected studies, only 1 case was reported to have a serious side effect (suspected osteonecrosis in one patient re-irradiated with CK) [15]. Yet, the lack of long-term data precludes a reliable assessment of potential stochastic effects. This underscores the critical need for standardized, prospective studies to better understand CK’s role in pediatric oncology.

In addition, Paddick et al. [20] measured extracranial doses from Gamma Knife Perfexion (GKP) intracranial stereotactic radiosurgery and modeled the malignancy risk from different treatment platforms. Doses were measured for 20 patients at distances of 18, 43, and 75 cm from the target, corresponding to the thyroid, breast, and gonads, respectively. Comparative data from other radiosurgery platforms were collected from the literature, and the National Cancer Institute RadRAT calculator was used to estimate excess lifetime cancer risk for different age groups. Results showed extracranial doses for GKP were 0.04%, 0.008%, and 0.002% of the prescription dose at 18, 43, and 75 cm, respectively. GKP had the lowest extracranial dose compared to linacs with micro-multileaf collimators (mMLC), linacs with circular collimators (cones), and CK. Estimated lifetime risks of radiation-induced malignancy were 0.03–0.88% for GKP, 0.36–11% for mMLC, 0.61–18% for cones, and 2.2–39% for CK [20]. This finding highlights a critical area of concern: the potential for increased risk of second tumors. In fact, this analysis underscores the importance of cautious application and rigorous long-term follow-up in pediatric patients treated with CK, reflecting the broader need for a balanced consideration of risks and benefits in employing this technology. In addition, while this study highlights the dosimetric advantages of the Gamma Knife, it is important to note that this treatment modality is limited to selected intracranial indications.

## 4. Discussion

### Narrative

Our narrative review delves into the limited yet emerging evidence regarding the application of CK in pediatric oncology. It is noteworthy that all the studies analyzed pertained to patients with intracranial or head and neck tumors, where the minimal or absent organ motion obviates the need to leverage CK’s advantages in real-time target tracking. Not surprisingly, of all the studies analyzed, only one reported on the treatment of gliomas. In fact, CK stereotactic RT is generally best suited for well-delineated tumor lesions due to its reliance on precise imaging and highly conformal dose delivery. In the case of infiltrative tumors, such as gliomas, the diffuse nature of these lesions poses challenges for achieving optimal target definition and dose conformity. While CK has been used for specific cases of gliomas with limited infiltration or well-delineated regions requiring focal treatment, its application in these scenarios remains limited. The findings across the reviewed publications indicate a potential for CK’s safety and efficacy in treating pediatric patients, with a rare report of severe complications like bone necrosis in the context of re-irradiation [15]. Notably, the reported 92% 3-year local control rate in one study on the treatment of brain tumors using a margin-free technique [19] points towards the potential benefits of CK in achieving effective tumor control with precise radiation delivery.

However, the current landscape of evidence, predominantly comprised of single-case reports and small case series, underscores the nascent state of knowledge regarding CK’s application in pediatric patients. The absence of large-scale, prospective clinical trials and the variability in reported outcomes highlight significant gaps in our understanding of CK’s long-term safety and efficacy. These limitations necessitate a cautious interpretation of the findings and a careful consideration of CK’s role in pediatric oncology.

CK offers several advantages that are particularly appealing in the treatment of pediatric tumors, including its precise imaging and tracking capabilities, non-invasive nature, and potential for reducing the number of treatment sessions [23,24,25]. These features suggest that CK could minimize exposure to healthy tissues and improve patient comfort, especially important in pediatric care. However, these advantages are tempered by notable concerns and limitations.

In fact, CyberKnife has well-recognized technical limitations. It is particularly effective only for tumors up to 3 cm in diameter, although in some cases, tumors up to a maximum size of 6 cm can be treated, depending on their location and shape. Furthermore, the use of CK requires tumors to be well-delineated on imaging studies, such as MRI or PET-CT, to ensure precise targeting during treatment. Additionally, the use of CK delivered with conventional fractionation [19] poses significant challenges, including its impact on departmental activity due to prolonged machine occupancy times and the intrinsic inhomogeneity of dose distribution produced by CK.

Moreover, the high initial costs, the complexity of treatment planning, and the potential for longer treatment sessions with CK present practical challenges to its widespread adoption [26,27,28]. Finally, the precision of CK, while a strength, also introduces the risk of a “dose-bath” effect, wherein low-dose radiation is distributed to a larger volume of healthy tissue than with traditional radiation therapy approaches [29]. This aspect is particularly concerning in pediatric patients, whose growing tissues are more susceptible to radiation-induced damage and who have a longer lifespan during which radiation-induced secondary cancers could develop [30,31].

In fact, the study by Paddick et al. [20] highlights the risk of radiation-induced malignancy with CK, confirming the need for a better understanding of the risk-benefit ratio in using CK for pediatric patients.

Therefore, our review suggests that considering the risks of carcinogenesis, CK is a reasonable option in two specific cases: i) treatments or retreatments in patients with an unfavorable prognosis, where CK offers a lower risk of acute or subacute side effects that could negatively impact the patient’s quality of life; and ii) treatments or retreatments in patients with a favorable prognosis, where conventional irradiation at curative doses is associated with an unacceptably high risk of acute or late side effects.

Moreover, our review indicates that future studies should focus on better quantifying the advantages of CK in reducing non-stochastic toxic effects and exploring its potential for dose escalation to enhance local tumor control. Additionally, it is crucial to quantify the risks of stochastic radio-induced effects, including carcinogenesis and transmissible mutations. In fact, the study by Paddick et al. [20] points out a broad range of risks for second cancers and does not address the risk of transmissible genetic effects, highlighting an area in need of further research.

Given these considerations, the potential of CK in pediatric oncology should be explored cautiously. Future studies with long-term follow-up and comparative data with conventional treatments are essential to clearly delineate CK's safety profile and therapeutic value. Until such evidence is available, clinical decisions regarding CK use in pediatric patients should be made within a multidisciplinary context, weighing potential benefits against risks (Figure 2) and considering each patient’s unique clinical scenario. Specifically, CK for treating pediatric patients, particularly those with tumors having a potentially favorable prognosis, seems justifiable only when conventional techniques pose an unacceptable risk of adverse effects.

In conclusion, the therapies currently available for pediatric RT, including linac-based IMRT/VMAT [32], proton therapy [33,34,35,36,37,38,39,40,41,42,43,44,45,46,47,48,49,50,51], 4π RT [52,53,54,55,56,57,58,59,60,61,62,63,64,65,66,67,68,69,70] (an advanced technique that delivers radiation from nearly unlimited angles around the patient, maximizing dose conformity to the tumor while minimizing exposure to surrounding healthy tissues), and CK, have been studied with varying levels of depth. While IMRT/VMAT and proton therapy are supported by a relatively robust body of clinical evidence, research on 4π RT and CK is less extensive, particularly in the pediatric population. This disparity highlights the need for a more systematic evaluation of emerging techniques to determine their optimal role in treatment planning. At a minimum, comparative planning studies are crucial to identify which modality offers the best therapeutic advantage for specific cases, balancing precision, safety, and accessibility.

CK and 4π RT, with their advanced precision and sparing of healthy tissue, hold significant promise for treating complex or inoperable pediatric tumors. CK real-time tracking and frameless delivery reduce treatment times, while 4π RT non-coplanar beam arrangements allow for unparalleled dose conformity. However, both techniques present unique challenges, including the potential for a low-dose bath with CK and concerns about the integral dose in 4π therapy, which could increase the risk of secondary malignancies. Proton therapy remains a benchmark for minimizing long-term toxicities due to its sharp dose fall-off, but its limited availability and cost are notable constraints. Table 2 shows a brief comparison of the main differences between the techniques currently available or being introduced into clinical practice in the pediatric tumor setting.

Future studies should not only aim to quantify and compare these techniques’ clinical outcomes but also evaluate their dosimetric advantages through rigorous planning studies. Such research is essential to establish evidence-based guidelines for selecting the most appropriate technique tailored to individual pediatric patients, ensuring both efficacy and safety in long-term outcomes.

## 5. Conclusions

CK offers potential advantages in precision and non-invasiveness for pediatric oncology, suggesting a promising role in treating various cancers. However, the evidence for its use in pediatric patients is limited, necessitating careful consideration. Key challenges include high initial costs, longer treatment times, and the need for specialized expertise. In pediatric settings, cautious application of CK is necessary to avoid risks associated with low-dose radiation over large body volumes. Therefore, decisions on CK application must be carefully weighed, focusing on minimizing long-term risks to young patients. Future research is needed to expand our understanding of CK safety and efficacy in pediatric oncology and guide its informed and judicious use. A recent German initiative has the potential to significantly advance our understanding of CK's role in pediatric CNS lesions. By systematically collecting and analyzing long-term data, it could pave the way for optimizing treatment strategies, improving patient outcomes, and fostering evidence-based integration of CK into pediatric oncology practice [71].

## Figures and Tables

**Figure 1 curroncol-32-00076-f001:**
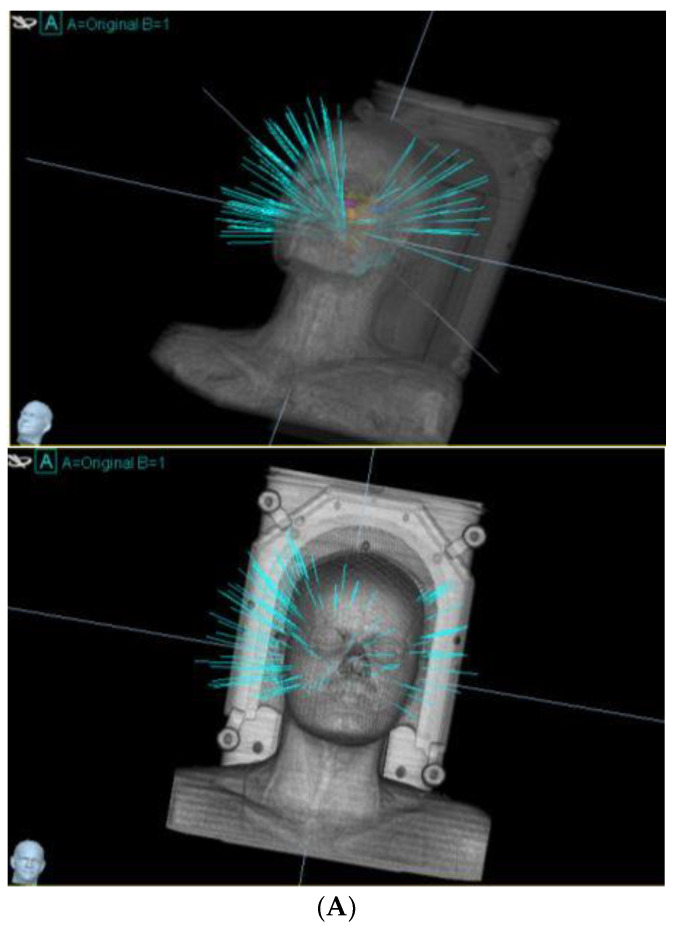
(**A**) Beam arrangement for CyberKnife treatment of a recurrent ependymoma in a young patient. (**B**) Dose distribution of the CyberKnife treatment.

**Figure 2 curroncol-32-00076-f002:**
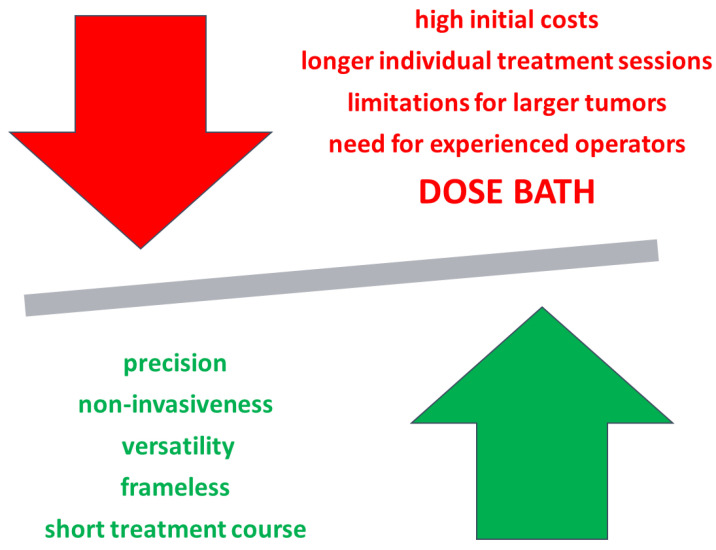
Advantages (green) and disadvantages (red) of CyberKnife. The “dose bath,” characterized by the delivery of low-dose irradiation to large body volumes, presents a persistent disadvantage for pediatric patients, especially when dealing with cases of favorable oncological prognosis. The potential risks associated with the “dose bath” may include long-term radiation-induced effects and an increased risk of secondary cancers, warranting careful consideration in treatment planning for this vulnerable population.

**Table 1 curroncol-32-00076-t001:** Summary of studies on CyberKnife in pediatric patients.

Authors/ Publication	Background/Aim Year	Material and Methods	Results	Conclusion
Harada et al./2000 [9]	To report a rare pediatric case of acoustic schwannoma with high proliferative potential.	Case study of a 10-year-old boy treated with subtotal resections and CyberKnife radiosurgery.	Rapid regrowth of the lesion, with immunohistochemical MIB-1 indices increasing from 2.3% to 14.7%.	Discussed proliferative potential of acoustic schwannoma. CyberKnife aided management.
Deguchi et al./2002 [10]	To present CyberKnife treatment of a 12-year-old boy with Juvenile Nasopharyngeal Angiofibroma.	Case report of CyberKnife therapy (3 treatments, 4512 cGy) after failure of external-beam radiation therapy.	Almost complete tumor disappearance after 7 months, no recurrence after 2 years.	CyberKnife is effective for Juvenile Nasopharyngeal Angiofibroma, offering an alternative to surgical and other radiotherapies.
Giller et al./2005 [11]	To report experience with CyberKnife in pediatric CNS tumors to avoid cognitive decline associated with other therapies.	21 children aged 8 months to 16 years underwent 38 CyberKnife treatments. Tumor types varied.	Local control achieved in pilocytic and anaplastic astrocytomas, some medulloblastomas, and craniopharyngiomas.	CyberKnife offers precise treatment for unresectable pediatric CNS tumors with no major complications.
Peugniez et al./2010 [12]	To assess the feasibility and tolerance of CyberKnife in children using conventional fractionation.	Report of 5 pediatric cases (ages 8–10) with recurrent brain tumors (optic pathway gliomas, pineal germinoma, Ewing sarcoma, and metastatic medulloblastoma) treated with CyberKnife.	Median treatment was 36.36 Gy over 20 sessions (31 days). No sedation or interruptions were required, and acute toxicity was minimal (grade 1).	Follow-up showed excellent tolerability with no severe toxicities. CyberKnife was a feasible and well-tolerated option for treating pediatric recurrences post-chemotherapy and prior radiation.
Li et al./2012 [13]	To report intracranial clear cell meningioma in two children and discuss the role of CyberKnife.	Two cases with clear cell meningioma, one receiving subtotal resection followed by CyberKnife.	Residual tumor shrank gradually post-CyberKnife in one case.	CyberKnife is a safe, effective adjuvant for clear cell meningioma, with NF2 gene mutation implicated in tumorigenesis.
Uslu et al./2013 [14]	To report CyberKnife use in optic nerve glioma treatment.	An 11-year-old girl with optic nerve glioma treated with CyberKnife fractionated stereotactic radiotherapy.	Marked tumor regression with no severe treatment-related toxicity after 1.5 years.	Supports further studies of CyberKnife for childhood optic nerve gliomas.
Gatz et al./2015 [15]	To report CyberKnife use in a multiply relapsed case of ameloblastic fibro- odontosarcoma.	15-year-old female treated with stereotactic CyberKnife reirradiation post-chemotherapy.	Complete remission maintained for 14 months post-reirradiation. Suspected bone necrosis.	CyberKnife combined with chemotherapy is effective in advanced ameloblastic fibro-odontosarcoma.
Nishimoto et al./2018 [16]	To report CyberKnife use in a malignant rhabdoid tumor the craniovertebral junction.	A 3-year-old boy treated with subtotal resection, CyberKnife radiotherapy, and chemotherapy.	Survived 29 months with local control but died of metastases.	CyberKnife is useful for local control in malignant rhabdoid tumor but requires multimodal treatment.
Mejías et al./2022 [17]	To report CyberKnife treatment of large brain metastases from Ewing’s sarcoma.	9-year-old boy treated with CyberKnife in two stages for brain metastases.	Complete resolution of lesions and good cognitive outcomes after 20 months.	Supports CyberKnife for large metastatic lesions in pediatric patients.
Fadel et al./2019 [18]	To review CyberKnife use in oculomotor nerve schwannomas.	Systematic review and two pediatric cases treated with fractionated CyberKnife radiotherapy.	Tumor control achieved without new deficits over 56–58 months.	CyberKnife is effective and well-tolerated for pediatric oculomotor schwannomas.
Mohamad et al./2020 [19]	To compare fractionated CyberKnife with IMRT in pediatric brain tumors.	52 pediatric cases treated with CyberKnife. Dosimetry compared with IMRT.	CyberKnife reduced normal tissue radiation volumes without compromising local control (3-year local tumor control: 92%).	Fractionated CyberKnife reduces irradiated tissue volume; results warrant prospective validation.
Paddick et al./2021 [20]	To measure extracranial doses and model malignancy risks from different SRS platforms.	Measured doses from Gamma Knife, linacs, and CyberKnife, modeling lifetime malignancy risks.	CyberKnife had highest extracranial dose and malignancy risk (2.2–39%).	Malignancy risk varies by platform; therapeutic reference levels proposed.
Yoo et al./2024 [21]	To evaluate CyberKnife for recurrent cranial medulloblastomas in pediatric and adult populations.	Retrospective review of 15 medulloblastomas in 10 patients treated with CyberKnife.	3-year local control: 65%, overall survival: 70%, progression-free survival: 58%. Better outcomes in pediatric patients.	CyberKnife is safe and effective, requiring tailored approaches for recurrence management.

**Table 2 curroncol-32-00076-t002:** Summary of main differences among available radiotherapy techniques in the treatment of pediatric tumors.

Aspect	Image-Guided Radiotherapy (IGRT)	Proton Therapy	Intensity-Modulated Proton Therapy (IMPT)	Gamma Knife	CyberKnife
Precision	Moderate (depends on imaging quality)	High (Bragg peak)	Very high (modulates intensity for optimized dose distribution)	Very high (designed for intracranial and small lesions)	Very high (real-time tracking
Impact on Surrounding Tissue	Moderate to high (depends on technique)	Minimal	Minimal (improved dose sculpting over standard proton therapy)	Minimal (steep dose fall-off within the cranium)	Minimal to moderate (CyberKnife offers very high precision with real-time tracking, reducing exposure to surrounding tissue. However, some low-dose scatter may occur, particularly for certain tumor locations.)
Treatment Duration	Weeks (fractionated)	Weeks (fractionated)	Weeks (fractionated)	Single session or a few sessions	1–5 sessions
Availability	Widely available	Limited	Very limited (requires specialized facilities and expertise)	Limited (dedicated for specific indications)	Quite available (depending on country)
Cost	Less expensive	Expensive	Very expensive (advanced technology required)	Expensive	Expensive (less expensive than proton therapy and IMPT, it is costlier than traditional image-guided radiotherapy).
Indication for Pediatric Use	Effective but higher exposure to non-target tissue	Excellent for minimizing long-term effects	Superior for highly complex or irregular tumors in sensitive areas	Excellent for intracranial tumors and small, well-defined lesions	Effective for irregular or moving tumors and situations requiring precision, No evidence suggesting higher risk of side effects compared to other high-precision modalities. The risk profile largely depends on tumor location and radiation dose.
Suitability for Large Tumors	Effective	Effective	Effective for both small and large tumors with complex geometries	Less suitable	Less suitable
Real-time Tumor Tracking	Limited (may include adaptive strategies)	Limited	Limited, but advanced planning compensates for movement	No	Yes
Special Applications	Broad use for various cancers	Tumors near critical structures	Complex, irregularly shaped tumors near critical structures	Small intracranial tumors	Irregular, small, or moving tumors
Dose Bath (Low-Dose Irradiation of Large Volumes)	High (large irradiated volumes due to less conformality)	Minimal (sharp fall-off with Bragg peak)	Minimal (improved dose sculpting reduces dose bath)	Minimal (confined to cranium)	High to moderate (low-dose scatter)
Sedation Requirement	Rarely needed (older children may remain still with immobilization devices)	Sometimes required for younger children due to long sessions	Often required for younger children due to precision and immobilization needs	Rarely needed (short treatment sessions)	Frequently required for younger children to ensure motion control during long-precise sessions
Grade of Evidence (Pediatrics)	High (extensive publications and clinical use in pediatrics)	High (extensive clinical evidence and recognized as pediatric-friendly)	Moderate to High (limited availability but growing evidence)	Moderate to High (well-documented for specific intracranial cases)	Low to moderate (well-documented, but pediatric-specific studies are fewer)

Abbreviations: IMPT: intensity-modulated proton therapy, GK: Gamma Knife, CK: CyberKnife.

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
