# Peer review of "CyberKnife in Pediatric Oncology: A Narrative Review of Treatment Approaches and Outcomes"

_curroncol, 2025, doi:10.3390/curroncol32020076_

Round 1
Reviewer 1 Report
Comments and Suggestions for Authors
The authors provide a narrative review on the use of CyberKnife irradiation in pediatric patients after having conducted a literature search which yielded 13 scientific papers on the subject.
While the authors have to be lauded for their intention to boost the use of CyberKnife SRS and SBRT in pediatric patients their review is a little irritating for it seems to consist of two parts with exclusive narratives.
While the Abstract, Introduction, Material & Methods and Results sections paint a rather rosy picture of everything CyberKnife, the Discussion stands in stark contrast. And if that was not enough the Discussion then veers off into totally new territory when a detailed comparison of CK to proton therapy, IMPT and 4π-RT (which is mostly experimental at this point) is introduced which has neither been mentioned in the title nor the abstract.
I strongly recommend re-working the first parts of the paper. It has to be clear that comparing normofractionated treatments with large margins within a curative setting with ultra-hypofractionated approaches to very small lesions as part of the treatment for recurrence is like comparing apples to oranges.
It is misleading to state: “Overall, despite their limitations, the selected studies seem to suggest that CK has a favorable toxicity profile concerning both acute and late deterministic effects” (line 191-192). Well, of course it is less toxic to treat lesions with a mean volume of 10cc but what conclusions are you suggesting here to draw from that fact? That we now start treating Glioblastomas with the CyberKnife? Or Medulloblastomas? Of course not. Because these tumors have to be treated with giant margins and of course these treatments are more toxic than treating a small meningioma or focal recurrence. Or are you suggesting that we now use ultra-hypofractionation for cranio-spinal axis irradiation? Of course not, I suppose. If you want to open up a proper comparison, compare modalities and treatment outcomes in situations where both or more approaches would have been possible (palliative treatment of small recurrences, grade 1 meningeomas etc.) But if you do not want to do that, abstain from making comments suggesting a superiority of Cyberknife over IMRT etc. without providing the proper context!
Now, the DISCUSSION in itself is the best part of the paper (because it provides detailed criticism) but seems a bit out of context. Abstract, Introduction and Results were all rather uncritical on Cyberknife bit now we are getting hypercritical? Where did that come from? Different authors? And then out of thin air the focus is on the comparison of CK to other techniques (like proton therapy or 4π RT (?!?)) without ever mentioning before that this was a huge part of this review? Weird. I don`t want to say that it is not worth reading but this section seems alien at this point. Who wanted to get the 4 π stuff in here?
CyberKnife is a great tool for some indications but we have to be realistic here, for most indications it is not. And that has to come across. Let the tone of the Discussion lead the way and cut all misleading statements.
So, when you cite somebody who did normofractionation on a Cyberknife (19), please also discuss critically that Cyberknife is not made for homogeneous dose distributions and critically discuss what the treatment times were as a result. And if that same author says the integral doses in the high and intermediate range (whatever that is) were in favour of the CyberKnife, please also cite how the low dose distribution was.
Or, when you want to insert a whole passage on GammaKnife (20) in this CyberKnife! review at least critically discuss that GK is only build for a very small number of indications inside the cranium.
Also
· Please repair both tables for the columns are touching in a way that there is nor space between the words of one column and the next
· Please say CK´s potential (line 29)…CK´s risk-benefit profile (line 37)…CK´s frameless design (line 80)…CK´s capacity (line 143)…CK`s potential (line 146)
· Please explain how there is a CBCT on a CyberKnife? (line 69)
Author Response
REVIEWER 1
Comment 1: The authors provide a narrative review on the use of CyberKnife irradiation in pediatric patients after having conducted a literature search which yielded 13 scientific papers on the subject. While the authors have to be lauded for their intention to boost the use of CyberKnife SRS and SBRT in pediatric patients their review is a little irritating for it seems to consist of two parts with exclusive narratives. While the Abstract, Introduction, Material & Methods and Results sections paint a rather rosy picture of everything CyberKnife, the Discussion stands in stark contrast. And if that was not enough the Discussion then veers off into totally new territory when a detailed comparison of CK to proton therapy, IMPT and 4π-RT (which is mostly experimental at this point) is introduced which has neither been mentioned in the title nor the abstract.
Response 1: Thank you for this insightful observation. We sincerely appreciate the feedback regarding the tone and structure of the manuscript. We have revised the tone of the Abstract, Introduction, Materials & Methods, and Results sections to present a balanced perspective on CK. Additionally, we have integrated comparative discussions of other technologies earlier in the manuscript for consistency.
In detail, we added to the manuscript the following sentences or paragraph:
Abstract:
- “the evidence remains limited, with the majority of studies involving small sample sizes and short follow-up periods”
- “Despite these advantages, CK is associated with significant limitations, including a higher potential for low-dose scatter (often referred to as the “dose-bath” effect), extended treatment times in some protocols, and high costs requiring specialized expertise for operation.”
- “This review emphasizes the need for balanced evaluations of CK and highlights the importance of planning prospective studies and long-term follow-ups to refine its role in pediatric oncology”
Introduction:
- “Despite these advantages, CK is associated with significant limitations, including a higher potential for low-dose scatter (often referred to as the “dose-bath” effect) and extended treatment times in some protocols, particularly when compared to other modalities such as Gamma Knife or proton therapy. Additionally, the high cost of CK systems and the specialized expertise required for their operation further restrict their widespread adoption, especially in resource-limited settings.”
- “predominantly consisting of small case series and retrospective reviews.”
- “The present review seeks to address this gap by critically evaluating the existing literature on CK application in pediatric patients. In doing so, the review aims to offer a balanced perspective, acknowledging both the potential advantages and significant challenges of CK in this sensitive population.”
Results:
- “Additionally, these studies do not provide sufficient data on the comparative effectiveness of CK versus other high-precision techniques like Gamma Knife, raising questions about its specific niche in pediatric oncology.”
- “For instance, the lack of comprehensive follow-up data on secondary malignancies remains a critical gap, particularly given CK’s association with a higher “dose-bath” effect compared to other technologies.”
- “While these results are promising, the study's small sample size and short follow-up highlight the need for further investigation into long-term outcomes and broader applicability.”
- “However, the study does not address the impact on broader clinical decision-making, particularly in the context of other available high-precision modalities.”
- “Nevertheless, these technical advantages must be weighed against practical challenges such as longer treatment durations, higher costs, and the need for specialized expertise.”
- “This underscores the critical need for standardized, prospective studies to better understand CK's role in pediatric oncology.”
Comment 2: strongly recommend re-working the first parts of the paper. It has to be clear that comparing normo-fractionated treatments with large margins within a curative setting with ultra-hypofractionated approaches to very small lesions as part of the treatment for recurrence is like comparing apples to oranges.
Response 2: Thank you for your comment. We hope the changes outlined in the previous response sufficiently address your concerns regarding the first part of the paper. To further clarify this topic, we have also added the following sentence at the end of the "Introduction" section:
- "Another aim of this review is to explore and discuss possible comparisons between the results of CK and those of other established techniques (such as image-guided RT, proton therapy, intensity-modulated proton therapy, and Gamma Knife) or emerging techniques (such as 4π RT), specifically within the clinical settings of pediatric tumors for which CK is intended, namely small tumor lesions treated with few fractions."
Comment 3: It is misleading to state: “Overall, despite their limitations, the selected studies seem to suggest that CK has a favorable toxicity profile concerning both acute and late deterministic effects” (line 191-192). Well, of course it is less toxic to treat lesions with a mean volume of 10 cc but what conclusions are you suggesting here to draw from that fact? That we now start treating Glioblastomas with the CyberKnife? Or Medulloblastomas? Of course not. Because these tumors have to be treated with giant margins and of course these treatments are more toxic than treating a small meningioma or focal recurrence. Or are you suggesting that we now use ultra-hypofractionation for cranio-spinal axis irradiation? Of course not, I suppose. If you want to open up a proper comparison, compare modalities and treatment outcomes in situations where both or more approaches would have been possible (palliative treatment of small recurrences, grade 1 meningiomas etc.) But if you do not want to do that, abstain from making comments suggesting a superiority of Cyberknife over IMRT etc. without providing the proper context!
Response 3: Thank you for your comment. Based on your suggestion we changed the sentence as follows:
- “Overall, despite their limitations, the selected studies suggest that CK has a favorable toxicity profile for both acute and late deterministic effects. However, this observation applies strictly to the specific clinical settings for which this technique is designed and potentially effective (namely, small tumor lesions treated with few fractions) and must be interpreted with caution given the absence of comparative studies”.
Comment 4: Now, the DISCUSSION in itself is the best part of the paper (because it provides detailed criticism) but seems a bit out of context. Abstract, Introduction and Results were all rather uncritical on Cyberknife bit now we are getting hypercritical? Where did that come from? Different authors? And then out of thin air the focus is on the comparison of CK to other techniques (like proton therapy or 4π RT (?!?)) without ever mentioning before that this was a huge part of this review? Weird. I don`t want to say that it is not worth reading but this section seems alien at this point. Who wanted to get the 4 π stuff in here?
Response 4: Thank you for your valuable feedback. We sincerely hope that the changes introduced in the abstract, introduction, and results sections have helped balance the tone of the review, aligning these sections more closely with the discussion. Additionally, we have removed the reference to the 4π technique from Table 2, focusing instead on comparisons with established and widely utilized techniques. References to the 4π technique have been retained only as a point of consideration, as it is presented as an ultra-precise technology but appears to share certain challenges with CK, particularly with respect to the dose-bath effect. We believe this adjustment improves the overall coherence and focus of the manuscript.
Comment 5: CyberKnife is a great tool for some indications but we have to be realistic here, for most indications it is not. And that has to come across. Let the tone of the Discussion lead the way and cut all misleading statements.
Response 5: Thank you for your comment. We hope that the changes made to the manuscript have effectively addressed the limitations of CK. Additionally, to further clarify this point, we have added the following sentences to the discussion section:
- "CyberKnife has well-recognized technical limitations. It is particularly effective only for tumors up to 3 cm in diameter, although in some cases, tumors up to a maximum size of 6 cm can be treated, depending on their location and shape. Furthermore, the use of CK requires tumors to be well-delineated on imaging studies, such as MRI or PET-CT, to ensure precise targeting during treatment.”
Comment 6: So, when you cite somebody who did normofractionation on a Cyberknife (19), please also discuss critically that Cyberknife is not made for homogeneous dose distributions and critically discuss what the treatment times were as a result. And if that same author says the integral doses in the high and intermediate range (whatever that is) were in favour of the CyberKnife, please also cite how the low dose distribution was.
Response 6: Thank you for your insightful comments. Based on your suggestions, we have revised the text as follows:
The sentence: “Results indicated that CK plans exposed significantly less normal tissue to high and intermediate doses compared to IMRT plans.” has been updated to:
- “Results indicated that CK plans exposed significantly less normal tissue to high doses (defined as ≥ 80% of the prescription dose or ≥ 40 Gy) and intermediate doses (defined as 80% > dose ≥ 50% of the prescription dose or 40 Gy > dose ≥ 25 Gy) compared to IMRT plans.”
In the Results, we have included the following clarification regarding reference 19:
- “Furthermore, the authors of this study did not report data on the volume irradiated at low doses, which may be correlated with the incidence of secondary malignancies.”
In the Discussion section, we have added the following sentence:
- “Additionally, the use of CK delivered with conventional fractionation [19] poses significant challenges, including its impact on departmental activity due to prolonged machine occupancy times and the intrinsic inhomogeneity of dose distribution produced by CK.”
Comment 7: Or, when you want to insert a whole passage on GammaKnife (20) in this CyberKnife! review at least critically discuss that GK is only build for a very small number of indications inside the cranium.
Response 7: Thank you for your comment. Based on your suggestion, we have added the following sentence to the end of the Results section:
- "In addition, while this study highlights the dosimetric advantages of the Gamma Knife, it is important to note that this treatment modality is limited to selected intracranial indications.”
Comment 8: Also, Please repair both tables for the columns are touching in a way that there is nor space between the words of one column and the next
Response 8: Thank you for your comment. We have modified the tables as per your suggestion to improve clarity
Comment 9: Please say CK´s potential (line 29)…CK´s risk-benefit profile (line 37)…CK´s frameless design (line 80)…CK´s capacity (line 143)…CK`s potential (line 146)
Response 9: Thank you again for your valuable feedback. All suggested corrections have been implemented as recommended.
Comment 10: Please explain how there is a CBCT on a CyberKnife? (line 69)
Response 10: Thank you very much for your clarification. Upon re-evaluating the literature, we confirmed that CBCT is used in combination with CK only in the context of CT angiography for the treatment of arteriovenous malformations, and not for oncological indications. As a result, we have removed the reference to CBCT from the text.
Reviewer 2 Report
Comments and Suggestions for Authors
Title of the manuscript: „CyberKnife in pediatric oncology: a narrative review of 2 treatment approaches and outcomes.” The manuscript is a review about the clinical application of cyber-knife (CK) in childhood malignancies. Since I have not found a very similar review in this topic, I recommend the consideration of the work for publication. However, I have some suggestions/concerns to the authors:
1, The title is about pediatric oncology, however all the cited articles elaborate on the CK treatment of brain tumors and head-and neck region cancers (where there is no real effect of organ movement). Please elucidate this finding of the work, explain the absence of extracranial CK RT in childhood malignancies (and consider the change of the title/abstract).
2, Considering brain tumors, I missed some explanation about the indication of CK stereotactic RT in case of infiltrative tumors, like gliomas.
3, In lines 64-65 the authors stated: “The intermittent 64 lower doses and lower dose rates permit healthy tissue recovery between fractions, thus 65 optimizing the therapeutic ratio.” I do not understand the role of this sentence after the introduction of CK application.
4, In line 75:” Within a few millimeters or within a millimeter?
5, One of the greatest values of the manuscript is the table about the comparison of different precision RT methods. I suggest modifying it to “fit to one page” with some clarifications and abbreviations.
6, What is the definition of IGRT? For me it is rather an online radiology-based control method without the designation of RT equipment. I recommend using linac-based IMRT/IMAT rather (or something similar).
7, I recommend some brief exposition of 4π RT method, since it is not familiar to the readers.
Author Response
REVIEWER 2
Title of the manuscript: „CyberKnife in pediatric oncology: a narrative review of 2 treatment approaches and outcomes.” The manuscript is a review about the clinical application of cyber-knife (CK) in childhood malignancies. Since I have not found a very similar review in this topic, I recommend the consideration of the work for publication. However, I have some suggestions/concerns to the authors:
Comment 1: The title is about pediatric oncology, however all the cited articles elaborate on the CK treatment of brain tumors and head-and neck region cancers (where there is no real effect of organ movement). Please elucidate this finding of the work, explain the absence of extracranial CK RT in childhood malignancies (and consider the change of the title/abstract).
Response 1: Thank you for your insightful comment, which provides an opportunity to clarify this aspect of our review. Based on your suggestion, we have made the following additions to the manuscript:
- Abstract: "All the studies analyzed reported cases of tumors located in the skull or in the head and neck region."
- Results: "However, all the cases and series analyzed focused on the treatment of tumors located in the intracranial or head and neck region."
- Discussion: "It is noteworthy that all the studies analyzed pertained to patients with intracranial or head and neck tumors, where the minimal or absent organ motion obviates the need to leverage CK's advantages in real-time target tracking."
We sincerely appreciate your suggestion, as it highlights an important finding of our review. However, we would prefer not to change the title, as our literature search was intentionally designed to encompass tumors across all anatomical locations. We believe the current title appropriately reflects the broad scope of our initial inquiry.
Comment 2: Considering brain tumors, I missed some explanation about the indication of CK stereotactic RT in case of infiltrative tumors, like gliomas.
Response 2: Thank you for your insightful comment. Based on your suggestion, we have expanded the discussion section to include the following explanation regarding the use of CyberKnife (CK) stereotactic RT in the treatment of infiltrative tumors, such as gliomas:
"Not surprisingly, of all the studies analyzed, only one reported on the treatment of gliomas. In fact, CK stereotactic RT is generally best suited for well-delineated tumor lesions due to its reliance on precise imaging and highly conformal dose delivery. In the case of infiltrative tumors, such as gliomas, the diffuse nature of these lesions poses challenges for achieving optimal target definition and dose conformity. While CK has been used for specific cases of gliomas with limited infiltration or well-delineated regions requiring focal treatment, its application in these scenarios remains limited."
We hope this addition clarifies the role of CK in the context of infiltrative tumors and strengthens the overall discussion on its indications.
Comment 3: In lines 64-65 the authors stated: “The intermittent 64 lower doses and lower dose rates permit healthy tissue recovery between fractions, thus 65 optimizing the therapeutic ratio.” I do not understand the role of this sentence after the introduction of CK application.
Response 3: Thank you for your observation. You are correct, and we agree that this sentence does not align clearly with the introduction of CK application. To address this, we have removed the sentence from the manuscript to improve clarity and maintain focus on the main topic. We appreciate your attention to this detail, which has helped streamline the presentation of the content.
Comment 4: In line 75:” Within a few millimeters or within a millimeter?
Response 4: Thank you for pointing out this ambiguity. To ensure accuracy and clarity, we have revised the text to specify: "within a millimeter." This adjustment reflects the precision typically associated with CyberKnife technology. We appreciate your attention to this detail, which has helped us improve the manuscript.
Comment 5: One of the greatest values of the manuscript is the table about the comparison of different precision RT methods. I suggest modifying it to “fit to one page” with some clarifications and abbreviations.
Response 5: Thank you for your positive feedback on the table comparing different precision RT methods. We appreciate your suggestion to improve its presentation. Based on your comment, we have modified the table to fit within one page by adjusting the formatting, abbreviating some terms, and ensuring all key information remains clear and concise. We believe these changes enhance the table's readability while preserving its informative value.
Comment 6: What is the definition of IGRT? For me it is rather an online radiology-based control method without the designation of RT equipment. I recommend using linac-based IMRT/IMAT rather (or something similar).
Response 6: Thank you for your comment. We appreciate your suggestion and have updated the manuscript accordingly. The term "IGRT" has been replaced with "linac-based IMRT/VMAT" to better align with its appropriate definition and context.
Comment 7: I recommend some brief exposition of 4π RT method, since it is not familiar to the readers.
Response 7: Thank you for your comment. To address this, we have added a brief explanation of the 4π RT method in the introduction section to provide context for readers who may not be familiar with it. The added text reads as follows:
"… an advanced technique that delivers radiation from nearly unlimited angles around the patient, maximizing dose conformity to the tumor while minimizing exposure to surrounding healthy tissues."
We hope this addition enhances the manuscript's accessibility for all readers. Thank you for highlighting this important point.
Reviewer 3 Report
Comments and Suggestions for Authors
The authors conducted a comprehensive narrative review on the utilization of CyberKnife in pediatric oncology. The evidence presented in this study is based on clinical studies, case reports, and reviews. Although the study includes relatively recent data, the majority of the evidence consists of single case reports. Since most of the oncology cases reviewed are CNS-based, I would suggest that the authors include Gamma Knife stereotactic radiosurgery in Table 2 for comparison.
Author Response
REVIEWER 3
Comment 1: The authors conducted a comprehensive narrative review on the utilization of CyberKnife in pediatric oncology. The evidence presented in this study is based on clinical studies, case reports, and reviews. Although the study includes relatively recent data, the majority of the evidence consists of single case reports. Since most of the oncology cases reviewed are CNS-based, I would suggest that the authors include Gamma Knife stereotactic radiosurgery in Table 2 for comparison
Response 1: Thank you for your valuable suggestion. In response to your comment, we have updated Table 2 to include a column on Gamma Knife stereotactic radiosurgery. This addition provides a comparison of the key features and applications of Gamma Knife with other radiotherapy techniques, including CyberKnife.
Round 2
Reviewer 3 Report
Comments and Suggestions for Authors
I am satisfied with the revision.